# DGPO: Mitigating Likelihood Displacement with Bidirectional KL Divergence Gap

## Abstract

The current margin-based model alignment method, represented by Direct Preference Optimization (DPO), aims to expand the margin between chosen and rejected responses. However, some works point out the log-probability of chosen response always decreases, thus affecting the likelihood of its generation. This likelihood displacement caused by gradient entanglement is a failure mode for preference optimization and has not been fully resolved. In this paper, we focus on forward and reverse Kullback-Leibler (KL) divergence on the probability distribution of preference pairs to form Divergence Gap Preference Optimization (DGPO). We prove DGPO can promote the increase of the chosen log-probability. Besides, DGPO also maintains a lightweight and automatic manner in real-world alignment. The downstream experimental results demonstrate that DGPO maintains competitive performance across various mainstream benchmarks without the reference model and additional key hyperparameters. Our code link is here.

## 1 Introduction

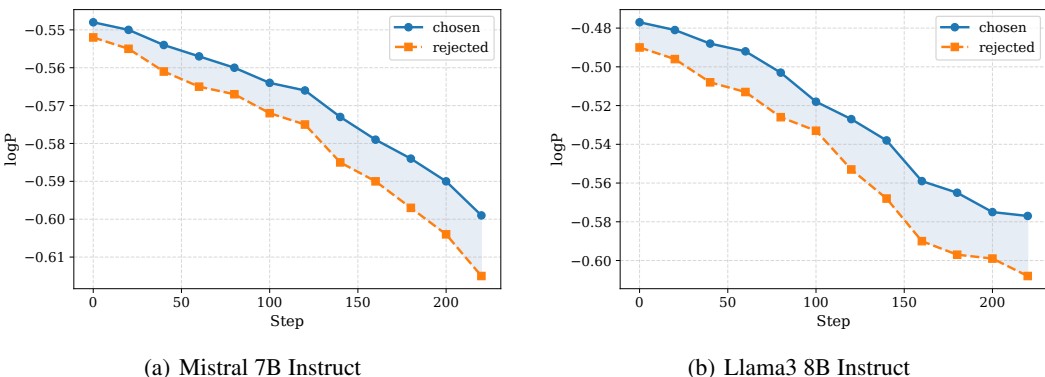

(a) Mistral 7B Instruct

(b) Llama3 8B Instruct

Figure 1: The chosen and rejected log-probabilities on the UltraFeedback for DPO.

Large language models (LLMs) typically undergo a pre-training stage on vast quantities of data to acquire general knowledge and patterns (Achiam et al., 2023). However, in practical applications, LLMs may generate inappropriate, harmful, or misleading content (Perez et al., 2022). By aligning LLM outputs with human intent, we can bring their responses closer to positive feedback, thereby better adhering to human values (Ziegler et al., 2019). InstructGPT (Ouyang et al., 2022) establishes Reinforcement Learning with Human Feedback (RLHF) as the predominant method for achieving alignment with human values. During the Proximal Policy Optimization (PPO) (Schulman et al., 2017) optimization process, the reward model evaluates the provided input and the resultant output, generating a reward score that serves as a feedback signal for reinforcement learning. Later, Direct Preference Optimization (DPO) (Rafailov et al., 2024) introduces a margin-based objective function. DPO directly incorporates human preferences into the loss function for model training, thereby eliminating the need for a separate reward model. Specifically, DPO achieves this by using the

log-probability margin between the policy model's responses and the reference model's responses as a proxy for the reward score. This approach effectively translates human preferences into an optimization criterion, allowing the model to be trained more efficiently and stably while reducing computational overhead.

As DPO is increasingly used for aligning LLMs (Tunstall et al., 2023; Ivison et al., 2023) , an abnormal phenomenon has been observed that the log-probabilities of chosen and rejected responses always exhibit synchronous decrease as shown in Figure 1, which is known as likelihood displacement (Razin et al., 2024). Although both log-probabilities decrease at different rates (Feng et al., 2024) and their margin is expanded, the decrease in the chosen log-probability is not what we want (Liu et al., 2020; Yuan et al., 2024b; Liu et al., 2024). Prior works (Yuan et al., 2024a; Razin et al., 2024) suggest that this situation is caused by the high correlation between chosen and rejected responses. This correlation leads to gradient entanglement during the training. It is mainly attributed to the fact that the current mainstream preference datasets (Cui et al., 2023; Bai et al., 2022) are generated with the assistance of LLMs, and there is a natural high similarity between chosen and rejected responses. In gradient entanglement, some margin-based baselines can also control response length bias (Azar et al., 2024; Meng et al., 2024), new parameters (Ethayarajh et al., 2024; Yang et al., 2025) and even complex function (Zhao et al., 2023b; Xu et al., 2024) to alleviate likelihood displacement to some degree but limit their real-world applications. Therefore, exploring how to improve the chosen log-probability with a more reasonable design is a key issue.

Inspired by this point, we introduce Divergence Gap Preference Optimization (DGPO), a principled solution that directly addresses this failure mode in a lightweight and theoretically rigorous manner. DGPO explores the core of gradient entanglement and leverages bidirectional KL divergence to seamlessly integrate two distinct optimization strategies. According to the theory of gradient entanglement, DGPO imposes more lenient constraints on the increase of the chosen log-probability. More importantly, DGPO serves as the most lightweight and automatic method for aligning LLMs without the need for additional costs in Table 1, which yields practical improvements.

Table 1: The costs on the margin-based objective. *Ref. free* and *No extra param.* indicate whether the alignment objective requires a reference model or additional hyperparameters other than $\beta$.

| Method | Ref. free | No extra param. |
|---------|-----------|-----------------|
| DPO | ✗ | ✔ |
| IPO | ✗ | ✗ |
| CPO | ✔ | ✗ |
| KTO | ✗ | ✗ |
| RRHF | ✔ | ✗ |
| SLiC-HF | ✔ | ✗ |
| SimPO | ✔ | ✗ |
| **DGPO** | ✔ | ✔ |

The contributions are summarized as follows:

a. DGPO leverages bidirectional KL divergences to facilitate the increase of the chosen log-probability. The forward and reverse KL divergences in the objective correspond to different optimization directions. We theoretically explain that DGPO guides the chosen and rejected probability distributions through distinct KL divergences to alleviate likelihood displacement caused by gradient entanglement.

b. DGPO is designed as a lightweight and automatic alignment method. Similar to some baselines without a reference model, DGPO saves GPU memory by 13.6% and reduces runtime by 25% compared to DPO. However, DGPO further eliminates the need for any additional hyperparameters beyond $\beta$. DGPO thus exhibits high applicability for practical deployment.

c. We perform extensive downstream comparisons with DPO and its prominent variants across multiple mainstream LLM evaluation benchmarks. DGPO maintains competitive performance on MT-Bench, AlpacaEval 2, Arena Hard, and Open LLM Leaderboard.

## 2 PRELIMINARIES

### 2.1 DPO AFTER SFT

The SFT stage generally pushes the model to assign high probabilities to chosen responses, essentially reaching a local optimal state for the likelihood of chosen responses to some degree. As a result, further increasing the chosen log-probability during the subsequent DPO stage becomes inherently challenging. In theory, any adjustment to the model's policy is likely to cause a slight decrease in the chosen log-probability. However, there is still significant room for reducing the rejected log-probability. To increase the margin, the model significantly reduces the rejected log-probability, while the chosen log-probability also slightly decreases.

In addition, for the pairwise preference dataset constructed based on the same prompt, the model may encounter the same prompt content multiple times within one epoch, especially when the preference pairs are very close. The gradient updates are repeated for the same context, and the gradient descent directions are very similar and continuously accumulate and amplify, which may lead to the model continuously reducing the log-probabilities beyond the necessary extent in the pursuit of expanding the margin, placing the model in an extreme state.

### 2.2 THEORY OF GRADIENT ENTANGLEMENT

Some works (Yuan et al., 2024a; Razin et al., 2024) have been considered the reason why the chosen and rejected log-probabilities usually decrease consistently during preference training. They both discover the correlation between chosen and rejected responses at the gradient level, which causes this anomalous phenomenon. Most of the current mainstream alignment methods can be formulated as a margin-based form:

$$\mathcal{L} = -(m(h_w(\log \pi_w) - h_l(\log \pi_l) + \Lambda(\log \pi_w))) \tag{1}$$

$$d_w := m'(h_w(\log \pi_w) - h_l(\log \pi_l))h'_w(\log \pi_w) + \Lambda'(\log \pi_w) \tag{2}$$

$$d_l := m'(h_w(\log \pi_w) - h_l(\log \pi_l))h'_l(\log \pi_l) \tag{3}$$

where $m$, $h_w$ and $h_l$ are scalar functions. $\Lambda$ is a scalar regularizer. On this basis, the changes in the chosen and rejected log-probabilities depend on their gradient inner product $\langle \bigtriangledown \log \pi_w, \bigtriangledown \log \pi_l \rangle$:

$$\triangle \log \pi_w \approx \eta \left( d_w \parallel \bigtriangledown \log \pi_w \parallel^2 - d_l \langle \bigtriangledown \log \pi_w, \bigtriangledown \log \pi_l \rangle \right) \tag{4}$$

$$\triangle \log \pi_l \approx \eta \left( d_w \langle \bigtriangledown \log \pi_w, \bigtriangledown \log \pi_l \rangle - d_l \parallel \bigtriangledown \log \pi_l \parallel^2 \right) \tag{5}$$

With Equations 4 and 5, the ideal conditions for the chosen log-probability to increase and the rejected log-probability to decrease are as follows:

$$\langle \bigtriangledown \log \pi_w, \bigtriangledown \log \pi_l \rangle \leq \frac{d_w}{d_l} \parallel \bigtriangledown \log \pi_w \parallel^2 \Longleftrightarrow \triangle \log \pi_w \geq 0 \tag{6}$$

$$\langle \bigtriangledown \log \pi_w, \bigtriangledown \log \pi_l \rangle \leq \frac{d_l}{d_w} \parallel \bigtriangledown \log \pi_l \parallel^2 \Longleftrightarrow \triangle \log \pi_l \leq 0 \tag{7}$$

However, when the chosen and rejected responses are highly correlated, due to the constraints of $\frac{d_w}{d_l}$ and $\frac{d_l}{d_w}$, this synergistic change effect on both log-probabilities becomes more pronounced (Pal et al., 2024; Tajwar et al., 2024). Combined with the explanation in Appendix A, this synergistic effect leads to the decrease in the chosen log-probability (likelihood displacement).

In the theory of gradient entanglement, other margin-based preference optimization loss functions can be expressed in a general form, as shown in Equation 1 of Table 2.

Table 2: The margin-based alignment reformulation

| Method | $m(a)$ | $h_w(a)$ | $h_l(a)$ | $\Lambda(a)$ |
|--------|--------|----------|----------|--------------|
| DPO | $\log\sigma(a - c_{ref})$ | $\beta a$ | $\beta a$ | - |
| IPO | $(a - (c_{ref} + \frac{1}{2\beta}))^2$ | $a$ | $a$ | - |
| CPO | $\log\sigma(a)$ | $\beta a$ | $\beta a$ | $\lambda a$ |
| KTO | $a$ | $\lambda_w\sigma(\beta a - (\log c_{ref}^w + z_{ref}))$ | $\lambda_l\sigma(\beta a - (\log c_{ref}^l + z_{ref}))$ | - |
| RRHF | $min(0, a)$ | $\frac{1}{\|y_w\|}a$ | $\frac{1}{\|y_l\|}a$ | $\lambda a$ |
| SLiC-HF | $min(0, a - \delta)$ | $a$ | $a$ | $\lambda a$ |
| SimPO | $\log\sigma(a - \gamma)$ | $\frac{\beta}{\|y_w\|}a$ | $\frac{\beta}{\|y_l\|}a$ | - |

**DPO ($\frac{d_w}{d_l} = 1$)** Increasing the value $\frac{d_w}{d_l}$ is the core and can provide more lenient conditions for the chosen log-probability increase.

**IPO/RRHF/SimPO ($\frac{d_w}{d_l} = \frac{\|y_l\|}{\|y_w\|}$)** When there is a length bias between the chosen and rejected responses ($\| y_l \| > \| y_w \|$), these methods provide lenient conditions.

**CPO/RRHF/SLiC-HF** The regularizer on the chosen responses increases $d_w$ without affecting $d_l$. Larger $\frac{d_w}{d_l}$ encourages the chosen log-probability to increase.

**KTO ($\frac{d_w}{d_l} \propto \frac{\lambda_w}{\lambda_l}$)** $\lambda_w$ and $\lambda_l$ are two additional hyperparameters. These two hyperparameters can regulate the constraints for the chosen log-probability increase.

## 2.3 EXPLORATION ON LIKELIHOOD DISPLACEMENT

We verify this problem at the whole sequence level. As shown in Figure 1, we use UltraFeedback (Cui et al., 2023) to fine-tune the models and find that the likelihood displacement is very severe. When we examine the current public preference datasets(e.g., UltraFeedback (Cui et al., 2023) and HH-RLHF (Bai et al., 2022)), we try Text-Embedding-3-Small [1] to calculate the embedding cosine similarity between pairwise responses. The result for UltraFeedback alone is 67.32% and that for HH-RLHF is surprisingly high at 90.16%! Given the same prompt, pairwise responses exhibit a high degree of similarity, which is attributed to the powerful language processing capabilities of LLMs.

Furthermore, we purposefully design meaningless text in the rejected response to explore whether likelihood displacement will be alleviated. We replace rejected response with other irrelevant datasets. At this point, the embedding cosine similarity drops to 12.85% and we observe to some extent opposite-directional changes in log-probability as shown in Figure 8. Likelihood displacement caused by the high embedding similarity of chosen and rejected responses is alleviated. However, when we return to reality, our focus is on how to design an alignment method for public preference datasets featuring high similarity to alleviate the decrease in chosen log-probability.

## 3 DIVERGENCE GAP PREFERENCE OPTIMIZATION (DGPO)

### 3.1 OBJECTIVE FUNCTION

LLMs alignment amounts to reshaping the relationship between the probability distributions over the chosen and rejected tokens. KL divergence is the most direct way to control these distributions. Because of its asymmetric feature, the forward and reverse KL divergence can be regarded as two distinct strategies. Our method exploits this design to guide the pairwise probability distributions. DGPO successfully alleviates likelihood displacement in the gradient entanglement framework.

**Forward KL divergence** The chosen distribution is the real probability distribution, and the other is the predicted distribution. We obtain the chosen response at this point. It is easy to understand that

---

[1]https://openai.com/index/new-embedding-models-and-api-updates

we need to expand the $D_{KL}(\pi_\theta(\cdot|x, y_w^{<i}) \| \pi_\theta(\cdot|x, y_l^{<i}))$ between the token probability distributions at each corresponding position in the chosen and rejected responses to weaken their correlation and push the rejected distribution away from the chosen distribution. $\pi_\theta(\cdot|x, y_w^{<i})$ and $\pi_\theta(\cdot|x, y_l^{<i})$ are the probability distributions of the chosen and rejected responses at the $i$-th token:

$$\theta_{w,i} := max(D_{KL}(\pi_\theta(\cdot|x, y_w^{<i}) \| \pi_\theta(\cdot|x, y_l^{<i}))) \tag{8}$$

**Reverse KL divergence**  The rejected distribution serves as the real probability distribution and the chosen is the predicted distribution. In this scenario, we can confirm that we have obtained the rejected response, but we are at a disadvantage in obtaining it. We aim to guide the chosen response by leveraging the existing rejected response as much as possible. Furthermore, the synergistic effect on the chosen and rejected log-probabilities reflects a comparable generation likelihood between the pairwise responses. So we aim at minimizing $D_{KL}(\pi_\theta(\cdot|x, y_l^{<i}) \| \pi_\theta(\cdot|x, y_w^{<i}))$ and enhancing their correlation. This strategy keeps both log-probabilities similar to inspire the chosen response from the rejected response:

$$\theta_{l,i} := min(D_{KL}(\pi_\theta(\cdot|x, y_l^{<i}) \| \pi_\theta(\cdot|x, y_w^{<i}))) \tag{9}$$

Finally, we equally integrate $\theta_w$ and $\theta_l$ to formulate sequence-level bidirectional KL divergence gap as follows. The detailed derivation process can be found in Appendix C:

$$\begin{aligned} L_{DGPO_1} &= -\log\sigma\left(\beta D_{SeqKL}\left(x, y, \pi_w \| \pi_l\right) - \beta D_{SeqKL}\left(x, y, \pi_l \| \pi_w\right)\right) \\ &\Leftrightarrow -\log\sigma\left(\beta(\log\pi_\theta(y_w|x) - \log\pi_\theta(y_l|x))\right) \\ &= -\log\sigma\left(\beta(\log\pi_w - \log\pi_l)\right) \end{aligned} \tag{10}$$

Here it can be clearly seen that this standard margin-based form of DGPO. However, DGPO needs to perform stable policy updates implicitly given the absence of the reference model and additional hyperparameters. We further apply an adaptive weight $(\pi_w + \pi_l)$ in $L_{DGPO_1}$:

$$L_{DGPO_2} = -\log\sigma\left(\beta(\pi_w + \pi_l)(\log\pi_w - \log\pi_l)\right) \tag{11}$$

- When the model has a high probability of both chosen and rejected responses for a certain prompt, it indicates that as $(\pi_w + \pi_l)$ increases, the model will increase its intensity to expand the margin.

- When the model has a low probability of both the chosen and rejected responses for a certain prompt, it indicates that the model's chosen and rejected responses probability estimates for this prompt are unreliable, and may have deviated from the initial reasonable distribution. At this time, $(\pi_w + \pi_l)$ becomes smaller, and the model will weakly optimize this prompt to avoid further distribution shift.

## 3.2 How does DGPO work?

Considering the Equations 6 and 7, $\frac{d_w}{d_t}$ is the important factor which can control more lenient condition to increase the chosen log-probability. DGPO extend the original $h_w(\log\pi_w)$ and $h_l(\log\pi_l)$ to $h_w(\log\pi_w, \log\pi_l)$ and $h_l(\log\pi_w, \log\pi_l)$. Here $h_w(a, b) = \beta(e^a + e^b)a$ and $h_l(a, b) = \beta(e^a + e^b)b$. Different from other baselines, $h_w(\bullet)$ and $h_l(\bullet)$ in DGPO objective function both are binary functions. Assuming $\log\pi_w$ and $\log\pi_l$ are independent, $h'_w(\bullet)$ is the partial derivative of $\log\pi_w$ and $h'_l(\bullet)$ is the partial derivative of $\log\pi_l$ for Equations 2 and 3.

> **Core**
>
> $$\frac{d_w}{d_l} = \frac{\frac{\partial h_w}{\partial a}}{\frac{\partial h_l}{\partial b}} = \frac{e^a + e^b + ae^a}{e^a + e^b + be^b} = \frac{(a+1)e^a + e^b}{(b+1)e^b + e^a} \tag{12}$$
>
> Finally:
>
> $$\frac{d_w}{d_l} = \frac{(\log\pi_w + 1)\pi_w + \pi_l}{(\log\pi_l + 1)\pi_l + \pi_w} \tag{13}$$

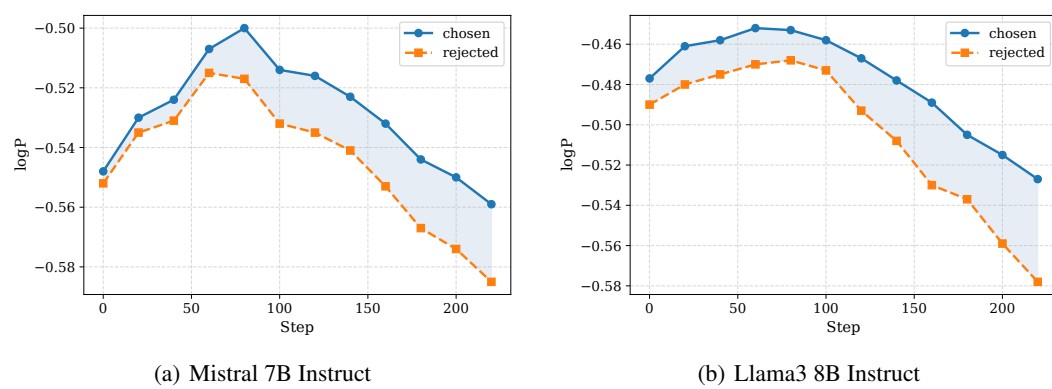

(a) Mistral 7B Instruct            (b) Llama3 8B Instruct

Figure 2: The chosen and rejected log-probabilities on the UltraFeedback for DGPO.

During the training process, different from DPO, DGPO observes the current $\log\pi_w$ and $\log\pi_l$. Actually $0 > \log\pi_w > \log\pi_l$ holds due to the inherent characteristic of margin-based alignment. Under this setting in Appendix D, we analyze the supported and violated conditions as follows:

**Supported Condition** $(0 > \log\pi_w > \log\pi_l > -1)$**.** It is easier for $\log\pi_w$ to increase and harder for $\log\pi_l$ to decrease. It is clear that $\frac{d_w}{d_l} > 1$ all the time for DGPO. As shown in Figure 2, DGPO provides a more lenient condition for the chosen log-probability increase.

**Violated Condition** $(-1 > \log\pi_w > \log\pi_l)$**.** At this time $1 > \frac{d_w}{d_l} > 0$ means it is harder for $\log\pi_w$ to increase and easier for $\log\pi_l$ to decrease.

We also conduct Mistral 7B and Llama3 8B on HH-RLHF. We show the chosen log-probability in Table 3. DGPO can still encourage the chosen log-probability increase.

Table 3: The chosen log-probability on HH-RLHF

| Step | 50 | 100 | 150 | 200 |
|---|---|---|---|---|
| Mistral 7B DPO | -1.22 | -1.23 | -1.26 | -1.30 |
| Mistral 7B DGPO | -1.16 | -1.12 | -1.09 | -1.06 |
| Llama3 8B DPO | -1.07 | -1.09 | -1.13 | -1.15 |
| Llama3 8B DGPO | -1.03 | -0.98 | -0.95 | -0.94 |

Notably, it can be seen in Section 2.2 that other margin-based methods rely on extra hyperparameters, content length bias or complex design to support the chosen log-probability increase. These settings limit the large-scale application of methods for mitigating likelihood displacement. In contrast, DGPO achieves a more reasonable and lightweight effect.

# 4 DOWNSTREAM PERFORMANCE

## 4.1 SETTING

Due to the significant correlation between the performance of preference optimization algorithms and upstream SFT models, we utilize the official Mistral 7B Instruct v0.2 (Jiang et al., 2023) and Llama3 8B Instruct (AI@Meta, 2024) as the initial SFT model for our comparative analysis. As the instruction-tuned variants, this versions exhibit robust and high-performance capabilities. We use UltraFeedback (Cui et al., 2023) as our preference dataset for the present study. We replicate DPO (Rafailov et al., 2024) and its variants IPO (Azar et al., 2024), CPO (Xu et al., 2024), KTO (Ethayarajh et al., 2024), RRHF (Yuan et al., 2023), SLiC-HF (Zhao et al., 2023b), ORPO (Hong et al., 2024) and SimPO (Meng et al., 2024). We use the recommended hyperparameters for all baselines

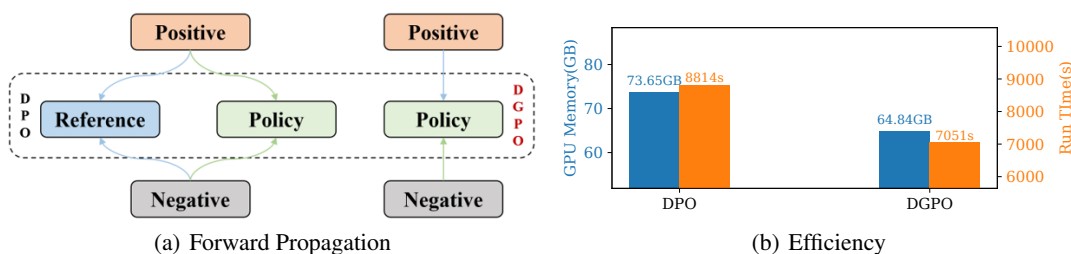

(a) Forward Propagation          (b) Efficiency

Figure 3: The efficiency of the Mistral 7B Instruct trained with DGPO and DPO. DGPO saves GPU memory by 13.6% and reduces run time by 25% compared to DPO.

and conduct distributed training using $8 \times$ NVIDIA A800 GPUs. The key baselines hyperparameters and more results about Llama3 8B Instruct can be been in Appendix E and F.

## 4.2 EFFICIENCY

As shown in Figure 3, it is reasonable to see that DGPO outperforms DPO in terms of computational efficiency (GPU memory and runtime). During the DPO alignment process, positive and negative feedback undergo a total of four forward propagation calculations through the reference model and policy model. However, DGPO does not require a reference model, thus requiring less GPU memory usage. Correspondingly, DGPO only needs to calculate two forward propagations of pairwise feedback in the policy model during training to reduce run time in each batch.

## 4.3 MT-BENCH

MT-Bench (Zheng et al., 2023) employs 80 high-quality two-turn questions to assess the answers produced by LLMs, utilizing GPT4 as the evaluation judge. To achieve more precise results, we adopt GPT4-1106-preview as the judge, superseding GPT4 in this benchmark. We use FastChat[2] (Zheng et al., 2023) to evaluate all the models. We present the results for each dialogue turn, including the average score and a radar chart, to comprehensively illustrate the performance of the models in Figure 4. In the initial dialogue turn, DPO demonstrates superior performance, surpassing other methods. However, its performance declines significantly in the subsequent turn. The radar chart further shows that DGPO exhibits outstanding performance in the Reasoning and STEM tasks. The overall results in Figure 4 and Table 7 obtained for DGPO consistently demonstrate exceptional performance across these two turns.

## 4.4 ALPACAEVAL 2

AlpacaEval 2 (Dubois et al., 2024) is an automatic evaluation based on LLMs, using the AlpacaFarm evaluation set (Dubois et al., 2023) for comprehensive assessment. LLMs and GPT4-1106-preview generate base responses and reference responses, respectively. These responses are then compared using an automatic annotator (GPT4-1106-preview) to ascertain the winning rate, which quantifies the performance of the LLMs. We use alpaca_eval [3] (Li et al., 2023) to evaluate all the models with 0.5 temperature. In Figure 5 both Win Rate and Length Controlled Win Rate, DGPO demonstrates improved results in comparison to DPO respectively. This observation in Figure 5 and Figure 9 suggests that, beyond fundamental abilities, DGPO has weaker performance in Length Controlled Win Rate. This is attributed to the lack of length normalization in DGPO.

## 4.5 ARENA HARD

Arena Hard (Li et al., 2024) contains 500 challenging questions filtered from data on the Chatbot Arena platform (Chiang et al., 2024), then performes automatic evaluation through GPT4. This benchmark combines community user feedback and automated evaluation to ensure the quality and

---

[2]https://github.com/lm-sys/FastChat
[3]https://github.com/tatsu-lab/alpaca_eval

| Method | 1st Turn | 2nd Turn | Average |
|--------|----------|----------|---------|
| SFT | 6.35 | 6.19 | 6.27 |
| DPO | 6.65 | 6.11 | 6.38 |
| IPO | 6.48 | 6.39 | 6.43 |
| CPO | 6.39 | 6.05 | 6.22 |
| KTO | 6.58 | 6.24 | 6.41 |
| RRHF | 6.42 | 6.60 | 6.51 |
| SLiC-HF | 6.44 | 6.48 | 6.46 |
| ORPO | 6.49 | 6.37 | 6.43 |
| SimPO | 6.60 | 6.38 | 6.49 |
| DGPO | 6.51 | 6.54 | **6.53** |

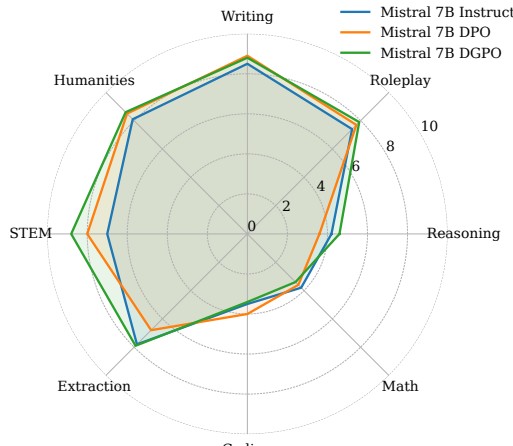

Figure 4: MT-Bench performance. Although DGPO does not achieve the highest rating in the initial evaluation round, it consistently maintains a high rating and demonstrates superior performance in the later round, ultimately culminating in the best overall result. Furthermore, across the eight tasks in MT-Bench, DGPO achieves notable improvement in the Reasoning and STEM tasks.

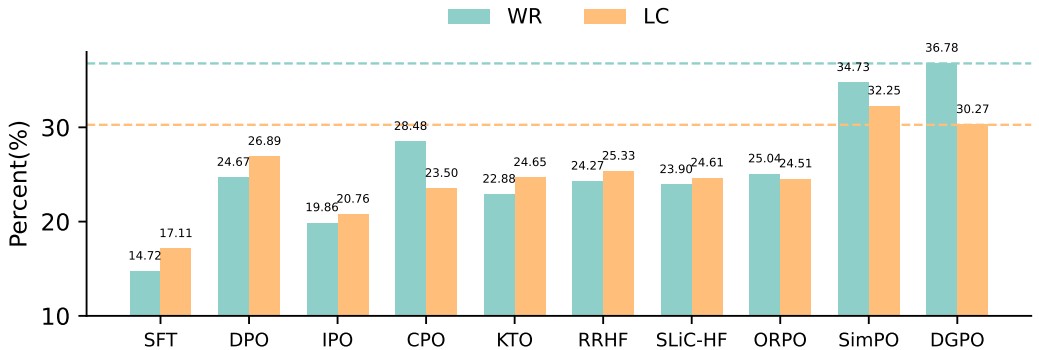

Figure 5: AlpacaEval 2 performance. DGPO demonstrates superior performance relative to other baselines in Win Rate. This indicates DGPO's effectiveness in generating responses that are well-aligned with expected outcomes. However, due to the lack of length normalization in DGPO, the improvement on Length Controlled Win Rate is not significant enough for SimPO.

diversity of evaluation data. We use arena-hard-auto [4] and take GPT4 as the judge. As show in Figure 6 and 10, DGPO have the optimal Win Rate and better 95% confidence interval.

## 4.6 OPEN LLM LEADERBOARD

Open LLM Leaderboard (Beeching et al., 2023) encompasses multiple sub-evaluation benchmarks, each employing distinct datasets (TruthfulQA (Lin et al., 2021), HellaSwag (Zellers et al., 2019), GSM-8K (Cobbe et al., 2021), WindGrande (Sakaguchi et al., 2021), ARC (Clark et al., 2018) and MMLU (Hendrycks et al., 2020)) to assess performance across various dimensions. These benchmark tests are designed to cover a broad spectrum of domains and tasks, and they evaluate the performance of LLMs by assessing its overall efficacy across all benchmarks collectively. We use lm-evaluation-harness [5] (Gao et al., 2024) to evaluate all the models on multiple benchmarks. In

---

[4]https://github.com/lmarena/arena-hard-auto
[5]https://github.com/EleutherAI/lm-evaluation-harness

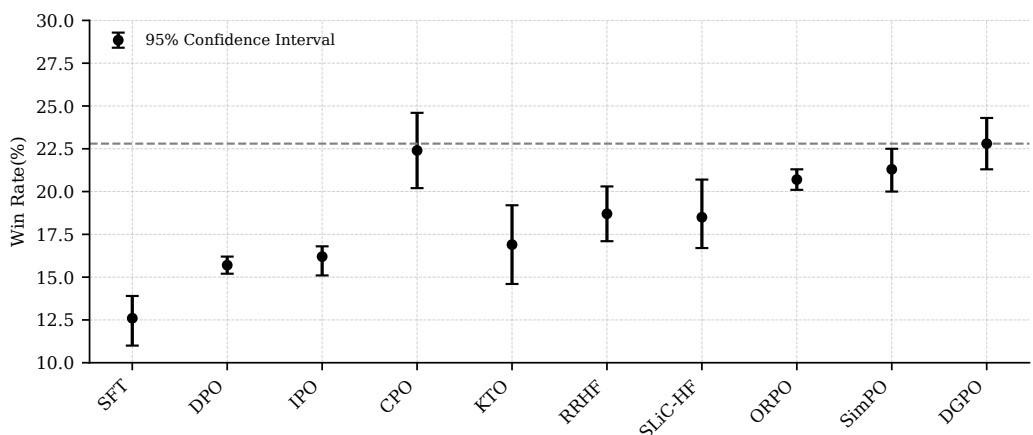

Figure 6: Arena Hard performance. The Win Rate of DGPO is optimal, and its 95 % confidence interval is relatively higher.

Table 4: Open LLM Leaderboard performance. DGPO outperforms other baselines overall.

| Method | TruthfulQA | HellaSwag | GSM-8K | WinoGrande | ARC | MMLU | Average |
|--------|-----------|-----------|--------|------------|-----|------|---------|
| SFT | 66.8 | 83.7 | 41.3 | 77.9 | 56.0 | 60.7 | 64.4 |
| DPO | 66.7 | 84.5 | 42.5 | 77.4 | 56.2 | 60.4 | 64.6 |
| IPO | 67.4 | 83.7 | 41.4 | 77.6 | 56.4 | 60.5 | 64.5 |
| CPO | 67.4 | 83.6 | 41.5 | 76.4 | 56.9 | 59.9 | 64.3 |
| KTO | 68.4 | 84.3 | 41.8 | 77.6 | 57.0 | 61.1 | 65.0 |
| RRHF | 68.1 | 84.6 | 38.2 | 76.6 | 60.5 | 59.8 | 64.6 |
| SLiC-HF | 65.3 | 84.1 | 40.0 | 76.9 | 58.4 | 60.0 | 64.1 |
| ORPO | 67.4 | 83.7 | 39.3 | 76.8 | 59.2 | 60.4 | 64.5 |
| SimPO | 69.0 | 84.1 | 40.3 | 78.7 | 59.4 | 60.5 | 65.3 |
| **DGPO** | 70.1 | 84.6 | 41.3 | 78.5 | 57.1 | 60.9 | **65.4** |

Table 4 and 8, DGPO takes optimal performance on some sub-evaluation benchmarks. Then DGPO is competitive in terms of average results.

## 4.7 ABLATION STUDIES

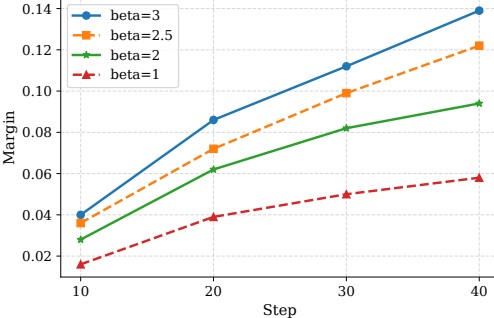

| $\beta$ | MT-Bench | WR(%) | LC(%) |
|---------|----------|-------|-------|
| 1 | 6.35 | 28.91 | 25.42 |
| 2 | 6.43 | 31.47 | 26.00 |
| **2.5** | **6.53** | **36.78** | **30.27** |
| 3 | 6.31 | 29.73 | 24.15 |

Figure 7: log-probability margin and performance on different $\beta$ values.

As shown in Figure 7, in the alignment objective function, the parameter $\beta$ serves as a temperature scaling factor, utilized to adjust the implicit reward margin. This parameter governs the extent of reliance on the behavior exhibited by the real distribution during the course of the optimization process. We employ various values of $\beta$ to train the model, with the aim of investigating its impact on the training outcomes. We also report the reward margin, utilizing $\beta$ values of 1, 2, 2.5, and 3. A clear trend is observed that the margin exhibits an increase as the value of $\beta$ is incremented and The best $\beta$ value is 2.5.

Within the DGPO, during the interaction between pairwise feedback, their respective optimization strategies ($\theta_w$ and $\theta_l$) remain independent of one another. In this section, we examine the impact of KL divergence term on the performance of DGPO. In Table 5 when employing a single optimization strategy, it is observed that the model maintains a high performance level similar to DGPO in the first turn of question answering on the MT-Bench. However, a significant decline in performance is evident in the second turn of question answering. For AlpacaEval 2, DGPO consistently maintains the highest win rate.

Table 5: Ablation studies performance on each term.

| Method | MT-Bench | | | AlpacaEval 2 | |
|---|---|---|---|---|---|
| | 1st Turn | 2nd Turn | Average | LC(%) | WR(%) |
| w/ $\theta_w$ | 6.42 | 6.14 | 6.28 | 25.86 | 27.47 |
| w/ $\theta_l$ | 6.50 | 5.90 | 6.20 | 28.38 | 25.64 |
| **DGPO** | **6.51** | **6.54** | **6.53** | **30.27** | **36.78** |

## 5 CONCLUSION

We present Divergence Gap Preference Optimization (DGPO), a lightweight method that mitigates likelihood displacement in margin-based preference optimization by exploiting a bidirectional KL divergence gap. Our theoretical analysis shows that DGPO refines the gradient entanglement condition to better support the chosen log-probability increase, and experiments across multiple benchmarks demonstrate its superior performance compared to strong baselines, while improving training efficiency. We believe DGPO provides a simple yet principled approach to more scalable and stable alignment.

## ETHICS STATEMENT

This work complies with the ICLR Code of Ethics. The data collection, usage, and model development processes comply with ethical standards regarding privacy.consent, and responsible Al practices. To the best of our knowledge, this study does not involve any data, methodologies, or applications that raise ethical concerns. The authors confirm that they have reviewed and followed the ICLR Code of Ethics throughout this research.

## REPRODUCIBILITY STATEMENT

To ensure the reproducibility of our work, for all theoretical results and corresponding insights claimed in this paper, we provide complete proofs and explanations in Sec.2.2, Sec.3 and App.A. In Sec.4.1 and App.E, we provide detailed descriptions of experimental setup. Then we have open-sourced our training code for the training process, which is available at https://anonymous.4open.science/r/DGPO-9DAB/. These efforts, combined with detailed descriptions throughout the paper, fully guarantee the reproducibility of our research findings and enable other researchers to validate and build upon our work.

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

## A    LIKELIHOOD DISPLACEMENT IN GRADIENT ENTANGLEMENT

Given the input prompt, LLMs generate responses token by token by mapping the token to logits, applying the softmax function to produce probability distributions and choosing the token with the highest probability. The probability distribution of the current output token depends on the previous tokens. Here we assume that the positive and negative feedback lengths are $L$ and the current token probability distribution is as follows:

$$\pi(y) = \prod_{i=1}^{L} \pi(y_i|y_{i<L}) \tag{14}$$

Under the theorem assumption of gradient entanglement (Yuan et al., 2024a), When the positive and negative feedback are the same from $1_{st}$ to $m - 1_{th}$ token in a sequence of length $L$, it is clear that the per-token chosen log-probability keeps unchanged with first-order Taylor expansion as follows:

$$\triangle\log\pi_w \approx 0 \tag{15}$$

Following the assumption (Pal et al., 2024), $\pi_{w,i}[j_i^*] \geq \pi_{l,i}[j_i^*]$ and $\pi_{w,i}[j] \geq \pi_{l,i}[j](j \neq j_i^*)$. When they are different on the $m_{th}$ token, the chosen log-probability increase as follows:

$$\triangle\pi_w(y_m|y_{<m}) \approx 1 + (\pi_{w,m}[j^*] - \pi_{w,m}[k^*]) \geq 0 \tag{16}$$

where $j^*$ and $k^*$ are the indices of $y_w[m]$ and $y_l[m]$ in the vocabulary. If the remaining tokens from $m + 1_{th}$ to $L_{th}$ are equal, the chosen log-probability will decrease as follows:

$$\triangle\pi_w(y_i|y_{<i}) \approx (1 - \pi_{w,i}[j_i^*])(\pi_{l,i}[j_i^*] - \pi_{w,i}[j_i^*]) - \sum_{j\neq j_i^*} \pi_{w,i}[j](\pi_{l,i}[j] - \pi_{w,i}[j]) \leq 0 \tag{17}$$

Therefore, the length of the same prefix has little effect on the log-probability. However, combined with Equation 4 and 5, it is clear that once the positive and negative feedback are highly similar and mostly clustered in the suffix, the log-probability of positive and negative feedback decreases synergistically, which is called likelihood displacement.

## B    UNRELATED NEGATIVE FEEDBACK

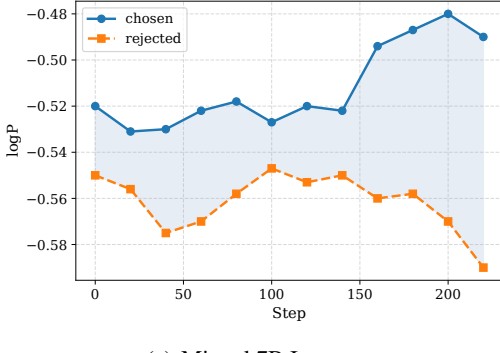
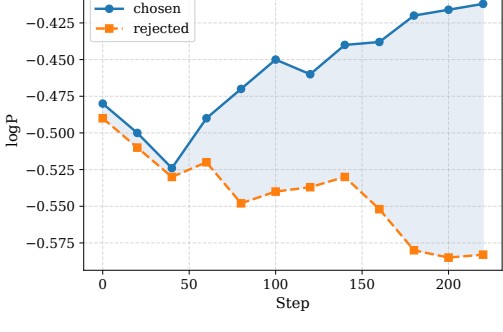

(a) Mistral 7B Instruct                    (b) Llama3 8B Instruct

Figure 8: The chosen and rejected log-probabilities with meaningless negative feedback for DPO.

## C DERIVATION

$$L_{DGPO_1} = -\log\sigma\left(\beta D_{SeqKL}(x, y, \pi_w \parallel \pi_l) - \beta D_{SeqKL}(x, y, \pi_l \parallel \pi_w)\right)$$

$$= -\log\sigma\left(\beta \sum_{i=1}^{n} D_{KL}(\pi_\theta(\cdot|x, y_w^{<i}) \parallel \pi_\theta(\cdot|x, y_l^{<i})) - \beta \sum_{i=1}^{n} D_{KL}(\pi_\theta(\cdot|x, y_l^{<i}) \parallel \pi_\theta(\cdot|x, y_w^{<i}))\right)$$

$$= -\log\sigma\left(\beta \sum_{i=1}^{n} \left(\log\frac{\pi_\theta(y_{w,i}|x, y_w^{<i})}{\pi_\theta(y_{l,i}|x, y_l^{<i})} - \log\frac{\pi_\theta(y_{l,i}|x, y_l^{<i})}{\pi_\theta(y_{w,i}|x, y_w^{<i})}\right)\right)$$

$$= -\log\sigma\left(2\beta \sum_{i=1}^{n} \left(\log\pi_\theta(y_{w,i}|x, y_w^{<i}) - \log\pi_\theta(y_{l,i}|x, y_l^{<i})\right)\right)$$

$$\Leftrightarrow -\log\sigma\left(\beta(\log\pi_\theta(y_w|x) - \log\pi_\theta(y_l|x))\right) \text{ (We put 2 into } \beta\text{)}$$

$$(18)$$

## D CONDITION

We still follow $a = \log\pi_w$ and $b = \log\pi_l$. Here we get $0 > a > b$. Then we set $f(x) = xe^x$:

$$f'(x) = (x + 1)e^x \qquad (19)$$

- $-1 > x$: $f(x)$ is monotonically decreasing.

- $x > -1$: $f(x)$ is monotonically increasing.

Therefore in Equation 12:

- $0 > a > b > -1$: $0 > ae^a > be^b$ and $\frac{d_w}{d_l} > 1$

- $-1 > a > b$: $0 > be^b > ae^a$ and $1 > \frac{d_w}{d_l} > 0$

## E SETTING

During the training process, we use Flash Attention 2 (Dao, 2023), a memory efficient and fast attention mechanism that significantly improves the speed of model training and inference by optimizing the computation process and memory usage. For DGPO and all baselines, we load the model with bf16 and conduct distributed training through Fully Sharded Data Parallel (FSDP) (Zhao et al., 2023a). We use AdamW (Loshchilov, 2017) optimizer and control the max length to 2048. Meanwhile We train one epoch on 8 NVIDIA A800 GPUs using 16 global batch size and 0.1 warmup ratio. We select recommended hyperparameters for each baseline, as shown in Table 6.

Table 6: Baselines hyperparameters.

| Baselines | Hyperparameters |
|-----------|-----------------|
| DPO | $\beta = 0.01$ |
| IPO | $\tau = 0.5$ |
| CPO | $\alpha = 1\ \beta = 0.01$ |
| KTO | $\lambda_w = 1\ \lambda_l = 1\ \beta = 0.01$ |
| RRHF | $\lambda = 1$ |
| SLiC-HF | $\beta = 1\ \lambda = 1$ |
| ORPO | $\lambda = 1$ |
| SimPO | $\beta = 2.5\ \gamma = 1$ |

# F    LLAMA3 8B INSTRUCT

Table 7: MT-Bench performance on Llama3 8B Instruct

| Method | 1st Turn | 2nd Turn | Average |
|--------|----------|----------|---------|
| SFT | 6.91 | 6.73 | 6.82 |
| DPO | 7.04 | 6.74 | 6.89 |
| IPO | 7.09 | 6.77 | 6.93 |
| CPO | 7.07 | 6.83 | 6.95 |
| KTO | 6.75 | 6.93 | 6.84 |
| RRHF | 6.72 | 6.84 | 6.78 |
| SLiC-HF | 6.95 | 6.65 | 6.80 |
| ORPO | 6.80 | 6.84 | 6.82 |
| SimPO | 7.16 | 6.68 | 6.92 |
| DGPO | 7.12 | 6.82 | **6.97** |

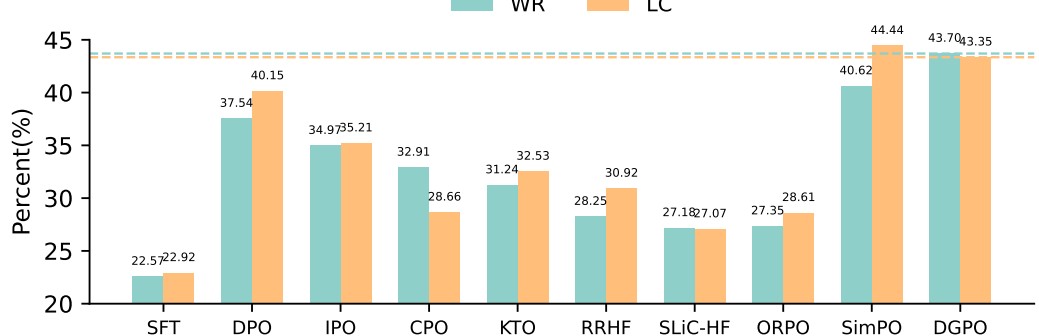

Figure 9: AlpacaEval 2 performance on Llama3 8B Instruct.

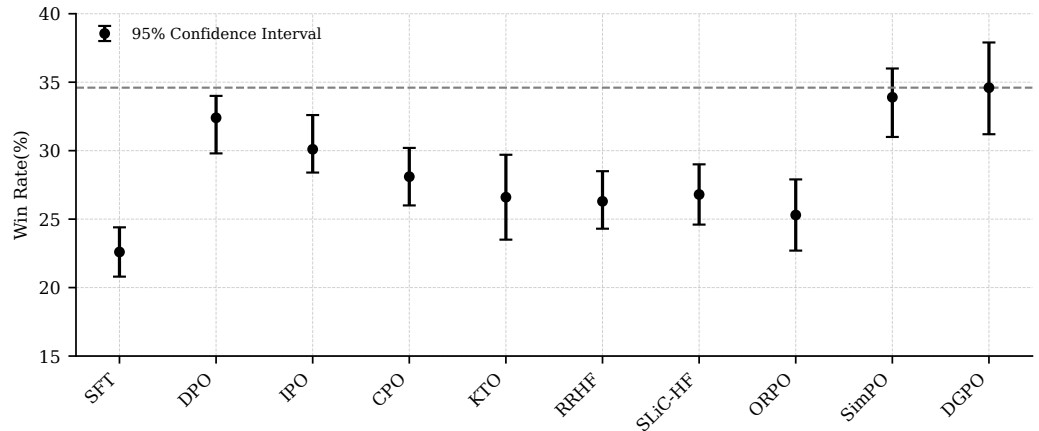

Figure 10: Arena Hard performance on Llama3 8B Instruct.

Table 8: Open LLM Leaderboard performance on Llama3 8B Instruct

| Method | TruthfulQA | HellaSwag | GSM-8K | WinoGrande | ARC | MMLU | Average |
|--------|-----------|-----------|--------|------------|-----|------|---------|
| SFT | 51.3 | 76.2 | 69.2 | 74.4 | 59.8 | 67.0 | 66.3 |
| DPO | 55.6 | 79.2 | 58.4 | 74.9 | 63.5 | 68.0 | 66.6 |
| IPO | 53.1 | 75.6 | 59.7 | 74.7 | 61.3 | 68.4 | 65.5 |
| CPO | 55.9 | 77.2 | 67.0 | 72.6 | 62.8 | 69.2 | **67.5** |
| KTO | 57.3 | 76.5 | 61.4 | 73.6 | 63.4 | 69.0 | 66.9 |
| RRHF | 52.5 | 78.9 | 67.3 | 73.6 | 61.6 | 67.3 | 66.9 |
| SLiC-HF | 52.0 | 77.4 | 68.1 | 73.5 | 61.2 | 69.7 | 67.0 |
| ORPO | 56.2 | 77.1 | 60.9 | 74.4 | 62.6 | 67.8 | 66.5 |
| SimPO | 57.5 | 78.3 | 60.0 | 73.8 | 62.6 | 68.5 | 66.8 |
| **DGPO** | 58.9 | 78.7 | 60.2 | 74.2 | 62.3 | 68.6 | 67.2 |

## G  RELATED WORK

**RLHF**  The core objective of aligning preferences for LLMs (Christiano et al., 2017) is to ensure that their behavior conforms to human values and ethical standards while minimizing erroneous outputs and biases. In RLHF, reinforcement learning techniques are combined with human feedback: evaluations of model-generated outputs are collected to train reward models. These reward models then guide the training of LLMs, optimizing the generated content to better align with human preferences. During the RLHF training process, the Bradley-Terry model (Bradley & Terry, 1952) is utilized to construct a reward function that assigns reward values based on how closely the model's outputs match human preference data. Reinforcement learning algorithms subsequently leverage this reward function to refine the model's behavior, thereby enabling it to generate outputs that yield higher rewards.

**DPO**  DPO (Rafailov et al., 2024) establishes a connection between the reward function and optimization strategy through the reparameterization of the reward model. This reparameterization facilitates a direct transition from reward optimization to strategy optimization, thereby enabling LLMs to achieve alignment using only simple classification loss. The approach not only obviates the need for an additional intermediate reward model and complex reinforcement learning algorithms during the alignment process but also establishes a theoretical framework for deriving strategies directly from human preferences.

## H  EXAMPLE COMPLETION

In Table 9, we use HuggingFace Pipelines[6] to present the answers given by Mistral 7B Instruct and Mistral 7B Instruct DGPO to a question from AlpacaEval 2 (Should I get my children a nanny? I'm so exhausted.). It can be seen that the model trained with DGPO provides more reasonable answers.

## I  LLM USAGE

The large language model (LLM) is used only to aid in polishing the writing and improving readability. The LLM does not generate research ideas, experimental designs, analyses, or conclusions.

---

[6]https://huggingface.co/docs/transformers/main_classes/pipelines

Table 9: Example Completion on AlpacaEval 2.

| Question | Should I get my children a nanny? I'm so exhausted. |
|---|---|
| **Mistral-7B-Instruct** | Deciding whether to hire a nanny is a personal decision that depends on various factors. Here are some things to consider:
1. Your budget: Hiring a nanny can be expensive, and you need to make sure you can afford it.
2. Your children's needs: Do your children require constant supervision or care? Do they have any special needs that a nanny could help with?
3. Your work schedule: If you work long hours or have an irregular schedule, a nanny could be a good solution to ensure your children are cared for when you can't be there.
4. Your family values: Consider your family values and whether having a nanny aligns with them.
5. Alternatives: Have you considered other options, such as a part-time babysitter, daycare, or family member?
Ultimately, the decision is yours, and it's important to weigh the pros and cons carefully before making a decision. If you do decide to hire a nanny, make sure to thoroughly vet them and ensure they are a good fit for your family. |
| **Mistral-7B-Instruct-DGPO** | Deciding whether to hire a nanny is a significant decision that depends on various factors beyond just your personal exhaustion. Here are some considerations to help you make an informed choice:
1. Budget: Hiring a nanny can be expensive, as you'll need to cover their salary, taxes, and benefits (if offered). You'll also need to provide them with a safe working environment and necessary equipment. Make sure you can afford the ongoing costs before making a commitment.
2. Childcare needs: Evaluate how much childcare you truly require. If you have young children with irregular schedules or special needs, a nanny might be an excellent investment. However, if your children attend school or daycare during the day, and you only need occasional help, other options like a part-time babysitter or a parent cooperative might be more cost-effective.
3. Family dynamics: Consider the impact on your family dynamics. Hiring a nanny can provide valuable support, but it also introduces a new person into your home. Make sure all family members are comfortable with the arrangement and understand their roles and responsibilities.
4. Quality of care: Research potential nannies thoroughly to ensure they have the necessary qualifications, experience, and references. A good nanny can provide high-quality, individualized care for your children, but a poorly matched or unqualified nanny could create more stress and chaos.
5. Alternatives: Explore other options before deciding on a nanny. Consider flexible work arrangements, splitting childcare duties with your partner, or enlisting the help of family members or friends. You might also look into community resources, such as parent cooperatives or local support groups, for additional assistance.
Ultimately, the decision to hire a nanny should be based on a thorough evaluation of your family's needs, budget, and priorities. If, after careful consideration, you determine that a nanny is the best solution for your situation, take the time to find a qualified and trustworthy individual who can provide the support and care your family needs. |

