# OpenReview forum: "DGPO: Mitigating Likelihood Displacement with Bidirectional KL Divergence Gap"
_ICLR.cc/2026/Conference — Submitted to ICLR 2026_

### Official Review · Reviewer_8rtp · 2025-10-27

**Soundness:** 2
**Presentation:** 1
**Contribution:** 2
**Rating:** 2
**Confidence:** 3

**Summary:**

The paper proposes a loss function named DGPO to mitigate the phenomenon of decreasing likelihood of positive samples observed in many preference alignment methods. The paper presents the empirical results with enough clarity, but the quality of the rest of the paper is poor (detailed in the weakness section). Although I appreciate the effort, I do not think the paper meets the acceptance bar of ICLR at its current state.

**Strengths:**

1. The paper presents the experimental settings and results clearly.

**Weaknesses:**

1. Low readability. The paper contains too many typos, grammatical errors, and notation ambiguities. The most serious one is the notation ambiguity in the proposed loss function (Equation 10). It is unclear what this loss function really means. The context seems to suggest that $\pi_w$ and $\pi_l$ represent $\pi(y_w)$ and $\pi(y_l)$. If so, then using $D_{KL}$ is an abuse of notation because $\pi_w$ and $\pi_l$ are not distributions but two likelihoods.
2. Questionable motivation. The paper situates itself as a method to alleviate the "problem" of the decreasing likelihood of positive samples. However, whether it is really a problem is still debatable in the first place. Literature arguing against this (such as [1]) is unfairly ignored by the paper.
3. Lack of meaningful baselines. Even if we assume that the suggested problem is indeed a problem, there are already a plethora of methods available, e.g., [2] can be a strong and meaningful baseline. However, the paper only compares the proposed method with relatively weak baselines that are not intended as cures to the decreasing likelihood of positive samples.

------
[1] Rafailov et al, 2024, From r to Q∗: Your Language Model is Secretly a Q-Function

[2] Chen et al., 2024, Noise Contrastive Alignment of Language Models with Explicit Rewards

**Questions:**

I do not have meaningful questions to ask besides the serious problems as I have listed in the weakness section.

---

> ### Author Response · Authors · 2025-11-16
>
> ### **Please refer to the Common Concerns Addressed.**
> ---
> ## W2
> We sincerely appreciate the reviewer’s insightful comment on the debatability of chosen log-probability decrease, and we would like to clarify our motivation on the practical background of SFT+DPO two-stage alignment pipeline:
>
> **The SFT stage generally pushes the model to assign high probabilities to chosen responses, essentially reaching a local optimal state for chosen response likelihood in some degree.** As a result, further increasing the chosen log-probability during the subsequent DPO stage becomes inherently challenging. In theory, any adjustment to the model’s policy is likely to cause a slight decrease in the chosen log-probability. However, there is still significant room for the rejected log-probability decrease. To increase the margin, the model significantly reduces the rejected, while the chosen also slightly decreases. This aligns with the observation in Figure 3 of Paper [1].
>
> In addition, for the pairwise preference dataset constructed based on the same prompt, the model may encounter the same prompt content multiple times within one epoch, especially when the preference pairs are very close. The gradient updates are repeated for the same context, and the gradient descent directions are very similar and continuously accumulate and amplify, which may lead to the model continuously reducing the log-probabilities beyond the necessary extent in the pursuit of expanding the margin, placing the model in an extreme state. Combined with the gradient entanglement theory, this indicates that chosen log-probability decrease may lead to a side effect of margin-based optimization in the SFT+DPO pipeline. **Therefore, This is also why we choose the LLM Instruction version for the preference pairs in the same prompt and encourage an increase in the chosen log-probability, reducing the risk of extreme solutions that may occur.**
>
> **We supplement all the above content in Section 2 of the revised version.**

---

### Official Review · Reviewer_x6EF · 2025-10-30

**Soundness:** 1
**Presentation:** 2
**Contribution:** 2
**Rating:** 2
**Confidence:** 4

**Summary:**

This paper proposes a new measure for preference optimization that weights \log\pi_w and \log\pi_l by the sum (\pi_w+\pi_l).

**Strengths:**

Gradient entanglement in DPO-like measures is an important problem.

**Weaknesses:**

Innovation:

All of the derivations in Section 2.1 have been presented more clearly in other papers; they do not need to be presented with this much detail in this paper.

Correctness:

Sections 3.1-3.2 use an incorrect definition of KL divergence.  KL divergence is defined to be the average log ratio between two different probability measures, averaged over all possible token sequences using weights given by the numerator measure.  Eq. (11) shows that you have misinterpreted D(pi_w||pi_l) to be the scalar \pi_w\log(\pi_w/\pi_l) --- KL divergence is defined to be the average of that quantity over all possible token sequences, not the scalar quantity computed based on a single scalar \pi_w.

The reasons given for maximizing Eq. (8) but minimizing Eq. (9) are post-hoc and incoherent.  In fact, the KL divergence from \pi_w to \pi_l is undefined, because both \pi_w and \pi_l are scalars, not distributions.

If h_w is a function of two arguments, then its scalar derivative h_w' is no longer well-defined; you must instead explicitly write dh_w/d\log\pi_w and dh_w/d\log\pi_l.  By making that substitution it is possible to recompute Eqs. (2) and (3) as attempted in Eqs. (12) and (13), but this recomputation is irrelevant, because the linearization in Eq. (4) and (5) is no longer true, so the conditions given in Eqs. (6) and (7) are no longer true.

**Questions:**

Presentation:

The derivations from Eq. (1) to Eq. (7) are interesting and have been covered in other papers, but they do not make obvious that Delta-log-piw < 0.  One way to make that obvious from these equations is to show that \Lambda'<0; there may be other interesting cases.

Minor presentation issues:

p. 2 both log-probability decrease -> the two log-probabilities decrease

high correlation between positive and negative feedback -> high correlation between winning and losing examples

Some works (Yuan et al., 2024a; Razin et al., 2024) have considered to be the reason why -> Some works (Yuan et al., 2024a; Razin et al., 2024) have considered the reason why

Eq. (1) is missing a close-paren.  Eq. (1) will be more readable if the symbol for loss is in calligraphic font, and if the symbol for log is in roman rather than italic font.

The presentation will be easier to follow if Eqs. (2) and (3) are moved after Eqs. (4) and (5) (as explanations of the terms d_w and d_l) rather than before.  Immediately after Eqs. (2) and (3) are presented for the first time, you should specify that prime denotes scalar derivative - this notation is common but not universal.  Immediately after Eqs. (4) and (5) you should specify that eta is the step size.

---

> ### Author Response · Authors · 2025-11-16
>
> ### **Please refer to the Common Concerns Addressed.**
> ---
> ## W1 Innovation:
> We hope this part can better help readers understand our work.
>
> ## W2 Correctness:
> Please refer to the Common Concerns Addressed. Then we clarify that the $h_w$ only focuses on $log\pi_w$ change in gradient entanglement. We assume $log\pi_w$ and $log\pi_l$ are independent although $h_w$ is binary function and $log\pi_l$ is constant for $h_w$ ($log\pi_w$ is constant for $h_l$). So $h_w^\prime(\bullet)$ is the partial derivative of $log\pi_w$ and $h_l^\prime(\bullet)$ is the partial derivative of $log\pi_l$. Then $d_w/d_l = \frac{\partial h_w/\partial a}{\partial h_l/\partial b} = \frac{e^a+e^b+ae^a}{e^a+e^b+be^b}$ in Eq. 12 of the revised version.
>
> **We supplement all the above content in Section 3.2 of the revised version.**
>
> ## Q1
> The derivations from Eq.1 to Eq.7 come from the gradient entanglement theory. The analysis is beyond the scope of our work.

---

### Official Review · Reviewer_ohV4 · 2025-11-01

**Soundness:** 3
**Presentation:** 2
**Contribution:** 2
**Rating:** 2
**Confidence:** 3

**Summary:**

The paper identifies a common failure in preference-based alignment called likelihood displacement, where training with DPO-like objectives lowers the likelihood of both the preferred and rejected responses when the two are very similar, and traces it to gradient entanglement in high-similarity preference data. To address this, it proposes DGPO (Divergence Gap Preference Optimization), a reference-free objective that compares the preferred and rejected responses using a bidirectional KL-divergence gap, effectively reweighting the gradients so the chosen response is more likely to increase in probability even when the pair is similar.

**Strengths:**

* The paper targets a concrete and currently observed problem in preference-based alignment (likelihood displacement) and clearly links it to gradient entanglement in high-similarity preference data.

* The core idea, using a bidirectional KL-divergence gap to reweight the chosen and rejected responses, is a principled and minimally invasive way to make it more likely that the chosen response’s probability actually increases.

**Weaknesses:**

* Could the authors also evaluate on Arena Hard benchmark, which is an extension of MTBench?

* In Figure2, could the authors also plot the DPO's chosen and reject probability to compare? It would also be nice to see the dynamics of the margin between the chosen and reject across the training. It is hard to see whether the method is effectively mitigating the decrease of chosen probability compared to DPO or other variants like SimPO (although its in Table 3, a visualization would be better).

* As an extension to previous comment, could we Table 3 only shows comparision of chosen. How about the rejected sentence?

* The performance gain seems very minimal compared to DPO and other variants.

* There are a few papers tackling the decrease of chosen probability during direct alignment. For example [1], [2]. The paper lacks comparison with such methods.

[1] https://arxiv.org/abs/2506.12725

[2] https://arxiv.org/abs/2405.16436

**Questions:**

See weaknesses above.

---

> ### Author Response · Authors · 2025-11-16
>
> ### **Please refer to the Common Concerns Addressed.**
> ---
> ## W1
> In fact, we have already conducted experiments on Arena Hard, and **the results are included in the original Section 4.5**.
>
> ## W2 & W3
> **Figure 1 in the original manuscript already contains the requested comparison of DPO’s chosen/rejected log-probability trends**. Besides, we first clarify a key misunderstanding: likelihood displacement is unrelated to the margin between chosen and rejected log-probabilities, but solely refers to the unintended decline of the chosen log-probability itself—as described in our manuscript (Section 1&2), even if the margin expands, likelihood displacement still occurs if the chosen log-probability drops.
>
> We appreciate your attention to the rejected log-probability in Table 3 and would like to further clarify the key points to avoid potential misunderstanding. Figure 2 in the original manuscript plots the rejected log-probability alongside the chosen log-probability. **Table 3 is designed as a supplementary empirical test to confirm DGPO’s ability to mitigate likelihood displacement (the chosen log-probability decrease). We originally omit rejected data here to avoid diluting the core conclusion.**

---

### Official Review · Reviewer_nUXw · 2025-11-01

**Soundness:** 3
**Presentation:** 3
**Contribution:** 2
**Rating:** 4
**Confidence:** 4

**Summary:**

This paper proposes Divergence Gap Preference Optimization (DGPO) to address the well-known issue of probability decrease in DPO, which is also termed "likelihood displacement".

The authors introduce a new reference-free objective function that effectively weights the standard DPO log-ratio margin by the sum of the chosen and rejected probabilities. This reduces the gradient signal when both probabilities are already low, preventing further displacement.

Theoretical motivation is provided via a simple "Bidirectional KL Divergence" theory and experiments are conducted on standard benchmarks.

**Strengths:**

**Timely Research Direction**

The focus on mitigating likelihood displacement via objective function reformulation is a valuable direction. I myself was recently considering a variant of this method and was glad to see the authors' exploration in this area, which produces a nice loss function addressing this.

**Practical Efficiency**

The proposed DGPO method is reference-free, reporting a ~13% GPU memory saving and 25% speedup over DPO. Though it must be said that in practice, the likelihoods can be computed offline (or asynchronously offline) and hence the memory savings of  not storing both \pi_theta and \pi_ref may be moot. Also, aren't such gains common to all reference free methods (like Simpo)?

**Addressing a Real Problem**

Targeting likelihood displacement is well-motivated. The intuition behind weighting the gradient by the sum of probabilities $(\pi_w + \pi_l)$ is a theoretically simple way to decrease weighting on the RL optimization when the model has already drifted too far from its initial high-likelihood region for a given prompt.

**Multi-turn Robustness**

DGPO appears to maintain performance better than DPO on subsequent turns in multi-turn benchmarks like MT-Bench (Figure 4), particularly in Reasoning and STEM tasks. This suggests the objective may be better for the model's general purpose capabilities than standard contrastive losses.

**Weaknesses:**

**1. Inconsistent gains against older baselines**
The empirical advantage over existing reference-free baselines (SimPO) is not clear-cut. In Appendix Figure 9 (Llama-3-8B), DGPO slightly beats SimPO on Length-Controlled Win Rate but loses to SimPO on raw Win Rate (43.3% vs 44.4%). Given that SimPO is a baseline from early 2024, a new method for ICLR 2026 should ideally demonstrate decisive improvements over it.

**2.1 Missing Literature on Reference Free DPO baselines**
Likelihood displacement, reference free DPO, and the related issue of model bias post DPO training have been active areas of research recently. The paper omits some recent works that also address these issues, sometimes with superior empirical results. For example, RefA [1] explicitly tackles length bias (a symptom of displacement) via reference-free token-level regularization. Game-theoretic approaches [3] and multi-preference optimized objectives [2] also mitigate these standard DPO failure modes, sometimes achieving higher win-rates.

**2.2 Outdated Performance Ceiling**
While DGPO improves over vanilla DPO, its absolute performance (~43% WR on Llama-3-8B) is significantly below the current state-of-the-art. The aforementioned recent methods [1, 2, 3] have achieved win rates between 50% and 60% on AlpacaEval 2 using similar base models and training data.

**Suggestion:** I recommend placing DGPO in the context of these more recent, and relevant baselines.

**3. Theoretical Derivation**
The connection between the Bidirectional KL theory and the final loss function needs more fleshing out, perhaps a rewrite to help me see it. Currently, it seems somewhat heuristic and can at best be termed a motivation rather than directly derived. The jump from minimizing forward/reverse KL to specifically weighting the margin by the scalar sum $(\pi_w + \pi_l)$ relies on simplifying assumptions that may not fully hold for complex sequence distributions. Can the authors consider a derivation in the style of the one taken up in the DPO paper. This would greatly strengthen their work.

---

Should these concerns be addressed, I would certainly consider raising my score.


**References**
[1] Gupta, T., et al. (2025). REFA: Reference Free Alignment with Fine-Grained Length Control. COLM 2025.
[2] Gupta, T., et al. (2025). AMPO: Active Multi Preference Optimization for Self-play Preference Selection. ICML 2025.
[3] Tang, X., et al. (2025). Game-Theoretic Regularized Self-Play Alignment of Large Language Models. arXiv preprint arXiv:2503.00030.

**Questions:**

**SimPO Comparison:** In your Llama-3-8B results (Figure 9), DGPO achieves a lower raw win rate than SimPO (43.3% vs 44.4%). Why does your method underperform the simpler SimPO baseline? Could the issue be related to noise on the benchmark, or lack of hyperparameter tuning?


### Minor Suggestion:

Please consider a language rewrite/polish. The paper could be polished to an even greater degree for readability.

---

> ### Author Response · Authors · 2025-11-16
>
> ### **Please refer to the Common Concerns Addressed.**
> ---
> ## Q1 SimPO Comparison
> Original Figure 9 actually shows DGPO achieves a higher raw Win Rate and a lower Length-Controlled Win Rate than SimPO.
> We have explained in original Section 4.4 that this is attributed to the lack of length normalization in DGPO.

---

### Author Response · Authors · 2025-11-16

# Common Concerns Addressed

## 1.Baselines(Reviewer nUXw W1&W2.1&W2.2, Reviewer ohV4 W4&W5, Reviewer 8rtp W3)
Thank reviewers for highlighting some works, and we clarify that our baselines selection is strictly grounded in DGPO’s core focus: **mitigating likelihood displacement caused by gradient entanglement in margin-based preference alignment, a theoretical framework we detail in Section 2.** We intentionally choose these methods as baselines because they all fit the standard margin-based alignment paradigm (Table 2) and process gradient entanglement condition in different ways. This ensures fair, in-paradigm comparisons: DGPO not only addresses gradient entanglement more reasonably but also retains lightweight design while matching their downstream performance.
These related works does not fit the "margin-based alignment + corresponding gradient entanglement condition mitigation" scope of our work. Their design paradigms and targeted mechanisms differ fundamentally from DGPO’s. So comparing them would not reflect DGPO’s meaning in solving the specific failure mode we focus on. **DGPO achieves a practical balance—dealing with gradient entanglement condition in a more reasonable way to mitigate likelihood displacement, simplifying deployment, and retaining competitive effectiveness rather than pursuing SOTA performance**.

## 2.Derivation(Reviewer nUXw W3, Reviewer x6EF W2, Reviewer 8rtp W1)
**We apologize for the over-simplification of some notations and derivation processes rather than factual flaws, leading to a misunderstanding.** $\pi_w$ and $\pi_l$ are actually $\pi_\theta(y_{w} |x)$ and $\pi_\theta(y_{l} |x)$ (the chosen and rejected response probability).

Next, we demonstrate the core derivation of DGPO:

The targets of these two strategies in DGPO are the token probability distributions for each corresponding position in the chosen and rejected responses. Hence, for Eq.8 and Eq.9 at the $i$-th token:

\begin{equation}
\theta_{w,i} := \max\left(D_{KL}\left(\pi_\theta(\cdot | x, y_w^{<i}) \parallel \pi_\theta(\cdot | x, y_l^{<i})\right)\right)
\end{equation}
\begin{equation}
\theta_{l,i} := \min\left(D_{KL}\left(\pi_\theta(\cdot | x, y_l^{<i}) \parallel \pi_\theta(\cdot | x, y_w^{<i})\right)\right)
\end{equation}

where $\pi_\theta(\cdot | x, y_w^{< i})$ and $\pi_\theta(\cdot |x, y^{< i}_{l})$ are the probability distributions of the chosen and rejected responses at $i$-th token. Then we integrate two strategies to form sequence-level KL divergence control:


\begin{equation}
\begin{aligned}
L_{DGPO_1} &= -\\text{log} \\sigma \left( \\beta D_{SeqKL} \left(x,y,\\pi_w \parallel \\pi_l \right) - \\beta D_{SeqKL} \left( x,y,\\pi_l \parallel \\pi_w \right) \right ) \\\\
&= -\\text{log} \\sigma \left( \\beta \\sum_{i=1}^{n} D_{KL}( \\pi_\\theta(\\cdot | x, y_w^{<i}) \parallel \\pi_\\theta(\\cdot | x, y_l^{<i})) - \\beta \\sum_{i=1}^{n} D_{KL}( \\pi_\\theta(\\cdot | x, y_l^{<i}) \parallel \\pi_\\theta(\\cdot | x, y_w^{<i})) \right ) \\\\
&= -\\text{log} \\sigma \left( \\beta \\sum_{i=1}^{n} \\left( \\text{log} \\frac{\\pi_\\theta(y_{w,i} | x, y_w^{<i})}{\\pi_\\theta(y_{l,i} | x, y_l^{<i})} - \\text{log} \\frac{\\pi_\\theta(y_{l,i} | x, y_l^{<i})}{\\pi_\\theta(y_{w,i} | x, y_w^{<i})} \\right) \right ) \\\\
&= -\\text{log} \\sigma \left( 2\\beta \\sum_{i=1}^{n} \\left( \\text{log} \\pi_\\theta(y_{w,i} | x, y_w^{<i}) - \\text{log} \\pi_\\theta(y_{l,i} | x, y_l^{<i}) \\right) \right ) \\\\
&\Leftrightarrow -\\text{log} \\sigma \left( \\beta ( \\text{log} \\pi_\\theta(y_w | x) - \\text{log} \\pi_\\theta(y_l | x) ) \right) \\text{(We put 2 into $\\beta$)}
\end{aligned}
\end{equation}

Here You can clearly see this standard margin-based form. **It is worth noting that DGPO needs to perform stable policy updates implicitly given the absence of the reference model and additional hyperparameters. We further apply an adaptive weight $(\pi_w+\pi_l)$ in Eq.11**:

1)When the model has a high probability of both chosen and rejected responses for a certain prompt, it indicates that as $(\pi_w+\pi_l)$ increases, the model will increase its intensity to expand the margin.

2)When the model has a low probability of both chosen and rejected responses a certain prompt, it indicates that the model's chosen and rejected response probability estimates for this prompt are unreliable, and may have deviated from the initial reasonable distribution. At this time, $(\pi_w+\pi_l)$ becomes smaller, and the model will weakly optimize this prompt to avoid further distribution shift.

**We supplement all the above content in the revised version.**

## 3.Typos(Reviewer nUXw Q2, Reviewer x6EF Q2, Reviewer 8rtp Q1)
Thank all the reviewers for providing valuable feedback on grammar errors and other issues. DGPO is elegant and we make unified modifications seriously in the revised version for better readability.

---

### Comment · Area_Chair_tP2g · 2025-11-28
**Please Check the Authors' Responses**

Dear Reviewers,

The authors have posted their responses. Could you please take a moment to review their responses and check whether your concerns have been adequately addressed (if you have not done it yet)? If possible, kindly initiate the discussion at your earliest convenience.

Your timely assistance is essential for keeping the review process on track. Thank you very much for your support and contribution.

Best regards, Your AC

---

### Author Response · Authors · 2025-12-02
**Summary Comment**

Dear Program Chairs, Senior Area Chairs, and Area Chairs,

We sincerely appreciate the time and effort you have invested in supervising the review process of our paper. Unfortunately, due to this bug in OpenReview, we are unable to have a thorough discussion with the reviewers. The gradient entanglement theory is an open problem. DGPO not only keeps good efficiency and achieves competitive results, but more importantly, provides a more reasonable solution to this problem. Our rebuttal has covered all original reviews, and **we appreciate you taking the time to carefully consider our work and correct the reviewer's misunderstanding of our work.**

If you still have some concerns, please leave them in the meta-review. In the camera-ready, we will further clarify any confusing wording. We thank the chairs for your professionalism and thoughtful feedback.

Best regards,

Authors of Submission 13993

---

### Meta-Review · Area_Chair_yu4s · 2025-12-31

**Summary:**

This paper studies the problem of likelihood displacement in preference-based alignment, where DPO-style objectives can decrease the likelihood of both chosen and rejected responses, especially when preference pairs are highly similar. Motivated by the observation that this issue comes from gradient entanglement, this work proposed Divergence Gap Preference Optimization (DGPO), a reference-free objective that reweights the standard DPO margin by the sum of the chosen and rejected probabilities, reducing harmful gradients when both likelihoods are already low.  Experiments on standard alignment benchmarks show modest but consistent improvements over vanilla DPO but gains over stronger recent baselines are limited.

The key strengths are (1) the problem of likelihood displacement in DPO-style alignment is important, (2) the idea of reweighting the DPO log-ratio margin by the sum of chosen and rejected probabilities is intuitive. The key weaknesses pointed out by reviewers are (1) the performance improvements are marginal and inconsistent, (2) the comparisons with recent reference-free and displacement-aware methods are missing, (3) the theoretical formulation is questionable, (4) the readability of the paper is not good, considering typos, grammar errors, notation ambiguities, and unclear derivations.

This paper received four negative ratings as initial recommendation. The rebuttal was provided while reviewers did not join the discussion. The area chair check both the rebuttal and the reviews. Although the gradient entanglement theory is an important and open problem, the major concerns and weaknesses on performance improvements, comparisons with recent methods, theoretical formulation and paper readability remain. For the selected baselines, the rebuttal claims that the baseline selection is strictly grounded in DGPO’s core focus, which limits the scope of the DGPO and does not provide the insights and conclusions compared with other types of preference alignment methods. For the theoretical formulation, the rebuttal provided a new detailed version, but still has readability issues, such as the definition of $\theta_{w,i}$ as the maximum value of KL divergence; also, it is not clear how the token-level KL divergence was transformed into a log-ratio of the chosen and rejected tokens. Considering the remaining concerns, the final recommendation is reject.

**Reviewer Concerns:**

For the selected baselines, the rebuttal claims that the baseline selection is strictly grounded in DGPO’s core focus, which limits the scope of the DGPO and does not provide the insights and conclusions compared with other types of preference alignment methods. For the theoretical formulation, the rebuttal provided a new detailed version, but still has readability issues, such as the definition of $\theta_{w,i}$ as the maximum value of KL divergence; also, it is not clear how the token-level KL divergence was transformed into a log-ratio of the chosen and rejected tokens. Considering the remaining concerns, the final recommendation is reject.

**Reviewer Scores:**

The above concerns are shared with different reviewers. As the key concerns remain, the final ratings would not be positive.

---

### Decision · Program_Chairs · 2026-01-26

Reject